# Two-Sided Matching Decision Method of Electricity Sales Package Based on Disappointment Theory

**Jianyu Ruan [1], Yingtong Wan [1] and Yuanqian Ma [2],***

1 Qixin Honors School, Zhejiang Sci-Tech University, Hangzhou 310018, China;
2020339930008@mails.zstu.edu.cn (J.R.); 2021339930016@zstu.edu.cn (Y.W.)
2 School of Information Science and Engineering, Zhejiang Sci-Tech University, Hangzhou 310018, China
* Correspondence: mayq666@zstu.edu.cn; Tel.: +86-134-380-26258

**Abstract:** Under the background of the "dual carbon" targets and continuously promoted power system reform, the application of a high proportion of renewable energy is becoming increasingly widespread. All sectors of society have greater demands for more appropriate electricity sales packages to guide the behavior of power users, which will in turn help conserve energy, reduce emissions, and finally achieve low-carbon operation of the power market economy. However, the existing methods of recommending electricity sales packages fail to provide appropriate and accurate recommendations for the users lacking preference information. Therefore, this paper proposes a two-sided matching decision-making method of an electricity sales package based on disappointment theory. First of all, according to the incomplete fuzzy preference relationship provided by the power user and the electricity sales package, the respective priority weight vector is calculated, and then the subjective satisfaction matrix of the power user and the electricity sales package is calculated. Next, the adjusted satisfaction matrix is calculated by adding the influence of the theory of elation and disappointment. Then, on the basis of the adjusted satisfaction matrix, an optimization model aiming at maximizing the total satisfaction of electric power customers and electricity sales packages is established, and the optimal stable matching model of electric power customers and electricity sales packages is obtained. Lastly, taking an industrial park in Zhejiang Province as an example, using the bilateral matching method proposed in this article, the optimal matching schemes for five electric power customers and six electricity sales packages is obtained, which shows the effectiveness and rationality of the two-sided matching decision-making method of electricity sales packages based on the disappointment theory.

**Keywords:** renewable energy; disappointment theory; incomplete fuzzy preference relationship; satisfaction matrix; two-sided matching; field survey

## 1. Introduction

In the context of the implementation of "dual carbon" targets, the reform of the power system is being promoted further, all sectors of society are paying more and more attention to the application of renewable energy, and the demand of electric power customers for clean energy is also growing. Not only China, but also the whole world is facing the dual carbon problem. EU member states are currently focusing on the Renewable Energy and Energy Efficiency Program [1]. Only by further increasing the proportion of renewable energy and making it deeply replace traditional fossil fuels can the high-carbon coal, natural gas, and other power systems develop in a low-carbon or even zero-carbon direction [2], eventually achieving a low-carbon economy. A large number of power-selling companies have emerged, and the demand for high-quality electricity from social electric power customers is also constantly increasing [3]. High-quality electricity refers to electricity with higher power quality indicators than those specified in existing public grid power supply regulations and restrictive standards [4]. Electricity sales companies need to attract

electricity users by setting reasonable electricity sales package prices, improving power quality, increasing the proportion of renewable energy, and providing additional value-added services. Electric power customers need to choose appropriate electricity sales packages according to different needs such as power supply reliability and electricity cost [5]. Therefore, the matching between electricity users and electricity sales packages can not only improve the revenue of electricity sales companies and meet the demand of users for high-quality electricity, but also have great significance in promoting the operation of the low-carbon economy.

At present, the main method for electric power customers to choose the electricity sales package is where the electricity sales company recommends the electricity sales package, and then the electric power customers make the decision. For power-selling companies, recommending appropriate power-selling package services for users is an effective way to improve the viscosity of electric power customers to power-selling companies and to attract a large number of new users [6]. At present, there are two main methods for recommending electricity sales packages: indirect recommendation and direct recommendation. There are two main categories of existing indirect recommendation methods, namely, statistical analysis recommendation and mathematical modeling recommendation. In statistical analysis, the main methods include regression analysis, cluster analysis, and preference analysis. The authors of [7] studied the impact of income, consumption expenditure, and price on household electricity consumption based on quantile regression analysis, avoiding sampling deviation, and providing a more accurate electricity sales package for families. The authors of [8] used time load and time series to cluster and analyze residential electricity customers, extracting customer behavior or load curves from the time series to recommend more targeted electricity sales packages to users. The authors of [9,10] used a preference analysis-based approach to develop fair energy allocation policies and unified pricing mechanisms for energy trading in P2P markets. However, most statistical analysis methods rely heavily on reliable and sufficient data. If the data quality is low or the quantity is small, it may reduce the accuracy of the recommendation results. In mathematical modeling, the main methods used are collaborative filtering and hybrid recommendation methods. The authors of [10,11] measured the relationship between users or products through collaborative filtering, looked for similar neighbor sets, and then completed the recommendation. However, the attribute of the item itself is not considered in the recommendation, which is likely to reduce the accuracy of the recommendation. The authors of [6], on the basis of the collaborative recommendation algorithm, introduced a recommendation method for electricity sales package in the Spark environment, comprehensively considered the electric power customers and electricity sales package volume for prediction and scoring, and obtained the recommendation data; The authors of [12] designed a recommendation system for electricity sales packages based on collaborative filtering and the hybrid Bayesian algorithm according to the electricity consumption characteristics of electric power customers. The authors of [13], on the basis of the artificial intelligence technology of collaborative filtering, recommended electricity sales packages according to the energy consumption characteristics of intelligent building customers. Although the indirect recommendation method is widely used, because the collaborative filtering method needs to cluster users, it needs to set the number of clusters in advance, which leads to low accuracy and efficiency of clustering and reduces the accuracy of the method. Since the model is based on historical data, the recommendation performance of the model may also be limited when there is a lack of historical data, or when the data were collected too long ago. The direct recommendation method is mainly applied through the use of iSelect [14], Check24 [15], and other power-selling package recommendation platforms. iSelect is an Australian company that provides comparison services for various products and services, including insurance, electricity, and gas providers, broadband plans, and financial products. It helps consumers compare different options and choose the best one according to their needs and preferences. Check24 is a German online comparison website that allows consumers to compare prices and services for a wide range of products and services. Check24 aims to provide

transparency and help customers make informed decisions by offering comparisons, reviews, and user ratings. However, this method is mainly based on the power-selling price and lacks many key factors that have an important impact on the electricity price; thus, the recommendation results are not accurate. In addition, with regard to the two major categories of recommendation methods mentioned above, when the power user and the electricity sales package evaluate each other, they do not take into account the situation that the power user may not know about the additional services of the electricity sales package, and that the electricity sales company may not know about the preferences of the power user. However, in the actual situation, due to various factors such as the source of information and the large number of electricity sales packages, it is often difficult for the power user to thoroughly understand the specific information of various electricity sales packages. Therefore, how electric power customers make the best choice under limited cognition is an urgent problem to be settled.

To sum up, this paper proposes a decision-making method for a user's electricity sales package considering an incomplete fuzzy preference relationship. This method can take the subjective psychological feelings into account, overcome the limitations of limited knowledge or understanding between the matching parties, and still accurately determine the optimal matching scheme under certain conditions of missing information. The process of this method is as follows: firstly, the priority vector of the electric power customers and electricity sales package is determined on the basis of an incomplete fuzzy preference relationship; then, a method to describe the satisfaction of electricity users and electricity sales packages according to disappointment theory is proposed; next, a multi-objective optimization model based on the two-sided matching method is proposed, which is aimed at maximizing the overall satisfaction of the matching between electric power customers and electricity sales packages; lastly, a case study of electric power customers in an industrial park in Zhejiang Province is demonstrated to verify the feasibility and effectiveness of the matching decision-making method of the user's electricity sales package considering incomplete fuzzy preference.

The characteristics of disappointment theory include its asymmetric sensitivity, dual-threshold effect, abstention effect, and intuitive effect. Asymmetric sensitivity refers to the fact that both matching parties are more sensitive to disappointment. The dual threshold effect refers to the existence of two thresholds for the sensitivity of both matching parties to risk. When one threshold is reached, the sensitivity to risk increases, and, when the other threshold is exceeded, the sensitivity to risk decreases. The abstention effect refers to the fact that both parties in the match will give up the option with higher returns due to fear of disappointment. The intuitive effect refers to the fact that, when faced with a large number of risks and decision-making choices, both matching parties often make choices on the basis of intuition. In practical bilateral matching problems, users may have expectations about certain packages, and they may be disappointed if these expectations cannot be fulfilled. Conversely, they may be delighted if certain packages exceed their expectations.

The characteristics of incomplete fuzzy preference are fuzziness, uncertainty, environmental dependence, relativity, and subjectivity. Ambiguity refers to the imprecise preferences given by both matching parties. Uncertainty refers to the hesitation phenomenon caused by the difficulty of predicting the consequences of each decision by both matching parties in decision making. Relativity refers to the fact that the decision results given by both matching parties will be influenced by the environment, resulting in discrepancies in the decision results. Subjectivity refers to the fact that every decision is the result of matching the subjective experiences and judgments of both parties.

In the electricity market, users usually have a limited understanding of electricity sales packages, and the knowledge and information they possess are also limited. In this case, users can only provide a vague general preference relationship, but cannot accurately express detailed preferences. For example, they might indicate a preference for a certain attribute as "high" or "moderate" rather than giving a specific numerical value. This fuzzy preference relationship can better reflect the cognitive limitations of users on electricity

sales packages. Furthermore, disappointment theory also plays a key role in this context. In the above text, we explained the concept of disappointment theory, whereby users have emotional experiences when faced with choices that may lead to disappointment. Users may have expectations for certain electricity sales packages, and they may be disappointed if these expectations cannot be realized. Conversely, they may be delighted if certain packages exceed their expectations. This psychological experience of disappointment and joy is one of the important driving factors behind user preferences. Combining fuzzy preference relations with disappointment theory can more accurately characterize user satisfaction. In the electricity market, due to the large number of packages, users are faced with the difficulty of making choices. The introduction of the bilateral matching method allows the system to comprehensively consider the user's fuzzy preference relationship and disappointment and joy psychology, so as to provide users with the best package matching. In this way, users do not need to face a complicated selection process, but can get recommendations for electricity sales packages that meet their vague preferences and lead to disappointment and joy, thereby improving their satisfaction.

## 2. Construction of Evaluation Index System for Electric Power Customers and Power-Selling Companies

### 2.1. Electric Power Customers' Evaluation Index of Electricity Sales Package

The power quality and power supply service of the electricity sales package are the two main considerations for electric power customers when selecting the electricity sales companies to launch different electricity sales packages.

#### 2.1.1. Clean Energy Ratio of the Package

"Carbon peaking" refers to the phenomenon where the total amount of carbon dioxide emissions reaches a historical peak at a certain point in time, during which there will still be fluctuations in the total amount of carbon emissions, but the overall trend is flat, and then the total amount of carbon emissions will gradually and steadily decline. "Carbon neutrality" refers to the phenomenon of neutralizing all carbon dioxide emissions, or even achieving negative carbon emissions, through fixed carbon emissions, afforestation, and other methods on the basis of "carbon peaking". Carbon peaking and carbon neutrality will have a huge impact on all aspects of our lives. The development of clean energy is an important means to help China achieve its carbon peak and carbon neutrality goals [16].

In 2003, the British energy white paper "Our Energy Future: Creating a Low-Carbon Economy" put forward the concept of a low-carbon economy for the first time. Since then, global energy has gradually entered a transitional stage toward a green and low-carbon energy structure. China has gradually promoted the development of clean energy into marketization, developing a clean energy structure system consisting of wind power, photovoltaic power generation, tidal power generation, hydropower, biomass power generation, and other clean energy sources [17].

According to data from the National Bureau of Statistics, the main power generation mode in China is still traditional thermal power generation. However, the proportion of clean electricity consumption in total energy consumption has increased year by year, from 20.5% in 2017 to 25.5% in 2021 [18], as shown in Figure 1.

In order to adapt to the accelerating adjustment of the global energy structure and the unchangeable trend of clean energy power generation, electricity sales companies need to increase the proportion of clean energy power generation in the package when formulating electricity sales plans and promote the development of "green electricity". At the same time, this will also become a crucial factor to electric power customers when choosing electricity packages. Now, we define the proportion of clean energy power supply to the total power supply in the electricity sales package as QJ. If $0 < QJ \leq 15$, then the proportion of clean energy is relatively small; if $15 < QJ \leq 25$, then the proportion of clean energy is moderate; if $QJ > 25$, then the proportion of clean energy is relatively high, and electricity users will also have a higher satisfaction with it [19].

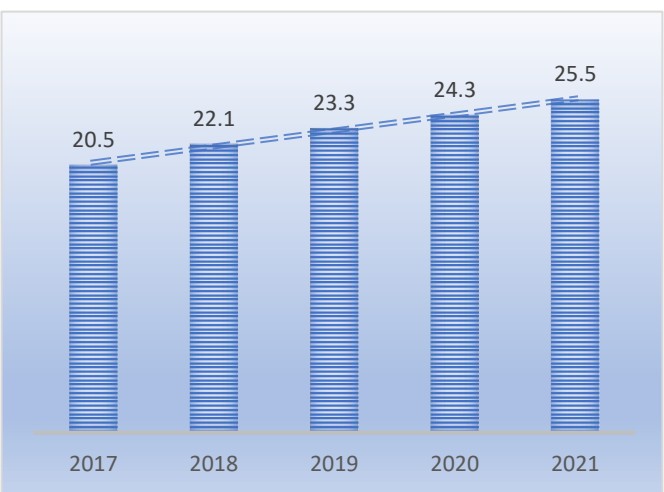

**Figure 1.** The proportion of clean energy consumption to total energy consumption from 2017 to 2021.

### 2.1.2. Power Quality

The factors that measure power quality mainly include voltage sag, harmonic, and three-phase voltage unbalance.

Voltage sag is a common voltage disturbance phenomenon in the distribution system. It is an accidental event. No matter how much the reliability of the power system is improved, voltage sag will still exist. Therefore, the degree of voltage sag and the frequency of voltage sag have become one of important standards to measure the quality of power supply by power supply companies. Voltage sag refers to the sudden event that the effective value of bus voltage drops sharply and rapidly and lasts for a very short time [19]. In the power grid, the duration of this phenomenon is mostly 0.5~1.5 s. The International Institute of Electrical and Electronic Engineers (IEEE) defines voltage sag as the phenomenon that the effective value of the supply voltage rapidly drops to 90–10% of the rated value and then returns to the normal value. Voltage sag is mainly caused by short-circuit faults, transformer excitation, and induction motor startup. Short-circuit faults mainly include three-phase grounding short-circuits, single-phase grounding short-circuits, two-phase interphase short-circuits, and two-phase grounding short-circuits. Voltage sag will lead to unit shutdown, misoperation of production machinery, machine damage, and other consequences, resulting in equipment damage, shutdown, or even scrap, and production line interruption. Its coverage is extremely extensive. So far, the semiconductor industry, petrochemical industry, automobile manufacturing, and chemical fiber industry have suffered huge human and material resources and financial losses caused by voltage sag [20]. Therefore, the smaller the amplitude and frequency of the voltage sag of the electric energy provided by the power-selling company, as well as the corresponding anti-interference core technology for different sensitive loads, the higher the power stability of the electricity sold by the company, and the more popular it will be with the electric power customers. A simple and effective voltage sag detection method was proposed in [21]. By sampling the signal for a period, the voltage amplitude is calculated from formula (1) to determine whether the voltage sag occurs.

$$U_{RMS} = \sqrt{\frac{1}{N} \int_{t_0}^{t_0+T} u_i{}^2(t)dt} \tag{1}$$

where $U_{RMS}$ is the effective value of voltage, $N$ is the number of sampling points in a period, and $T$ is the signal period.

Harmonic current refers to the current of each sinusoidal component whose frequency is an integral multiple of the frequency of the original periodic current when the non-

sinusoidal periodic current function is expanded in the Fourier series. With the rapid development of science and technology, a large number of powerful electronic nonlinear devices have emerged, such as computers, monitors, induction cookers, and washing machines [22]. There are also traditional nonlinear devices, such as transformers, rotating electrical machines, and fluorescent lamps [23], which are more and more widely used in low-voltage distribution networks and by ordinary electric power customers. Although the single harmonic impact of these new harmonic sources is small, due to their large number, the cumulative harm can not be ignored, which will cause serious harmonic pollution. The harm of harmonic current is mainly reflected in two aspects: first, harmonic current will cause power loss and increase the burden of users' electricity charges; second, the increase in current will cause the equipment temperature to rise, accelerate the insulation aging, and greatly shorten the service life of the equipment. Therefore, when formulating the power-selling plan, the power-selling company needs to consider the impact of harmonic current comprehensively and install some adaptive filters to reduce harmonic, thus reducing losses, reducing the burden of electricity charges of electric power customers, and protecting equipment and lines to a certain extent, which can effectively control the daily operation and maintenance costs [24]. The detection methods of harmonic active current, reactive current, and load harmonic current were described in detail in [25]. In the study of power grid power quality, the total harmonic distortion rate (THD) is generally used to characterize the harmonic level of the power grid, and the specific formula is as follows:

$$THD = \frac{\sqrt{\sum\limits_{h=2}^{\infty} U_h{}^2}}{U_1} \times 100\% \tag{2}$$

where $U_1$, $U_h$ are the amplitude of the harmonic voltage and the amplitude of the fundamental voltage.

Three-phase voltage unbalance refers to the inconsistent amplitude or phase angle of three-phase voltage. When a three-phase voltage unbalance occurs, negative sequence current and zero sequence current will be generated correspondingly, which will cause serious loss and great voltage drop to the power line, and interfere with the communication system, resulting in the heating, vibration, and loss of the rotating motor and transformer, and affecting the service life of the electrical equipment [26]. Its degree is characterized by three-phase voltage unbalance, which is generally calculated by the ratio of the effective value of the negative sequence component of voltage or current to the positive sequence component. Three-phase voltage imbalance has different degrees of impact on the power supply and distribution system, mainly on transformers, electrical equipment, and transmission lines. It will lead to an increase in transformer load loss, a reduction in overload capacity, an increase in core eddy current loss, and an intensification of heating, reducing the service life of the transformer. For ordinary electric power customers, three-phase load asymmetry will lead to the deviation of the neutral point, resulting in the problem of user voltage deviation, and leading to the unstable operation of electrical equipment. In the process of current transmission, the greater the three-phase unbalance, the greater the line loss, and the lower the economy of power transmission will be [27]. Therefore, when electric power customers choose power-selling companies, three-phase voltage unbalance is also an important consideration index. The lower its value, the more economical and high-quality the power supply will be. In a three-phase power system, three-phase unbalance is usually expressed quantitatively by three-phase voltage unbalance rate (PVUR):

$$PVUR = \frac{\max\{|V_A - \overline{V}|, |V_B - \overline{V}|, |V_C - \overline{V}|\}}{\overline{V}} \times 100\% \tag{3}$$

where $V_A$, $V_B$, $V_C$ respectively represents the A, B, C phase voltage, and $\overline{V}$ is the value of three-phase voltage.

2.1.3. Power Supply Service

The power supply service mainly consists of the ordinary power supply service and value-added service.

The so-called power supply service refers to the service that the power supply company provides customers with some corresponding valuable business activities in the form of labor services to enable customers to purchase electricity and meet their production and living needs. Due to the social, systematic, special, and developmental characteristics of power supply service and the imbalance between power supply capacity and user load, it is slightly different from the general service industry, which generally includes basic services such as meter reading and charging, fault repair, complaint reporting, inquiry, and consultation. The well-known foreign scholars Parasuraman, Zeithaml, and Berry built the service quality gap module in 1985, believing that service quality is the gap between customer expectations and customer experience, and they proposed 10 dimensions that affect perceived service quality and customer perceived service, namely, responsiveness, accessibility, security, reliability, ability, politeness, communication, credibility, understanding, and tangibility. The authors of [28] elaborated on the relevant contents of power supply service quality and clarified the connection and interrelation between the service of power supply companies and customers' expectations of service quality. In short, if the power supply company can sell electricity according to the four principles of "high quality, convenience, standardization, and sincerity", the higher the quality of electricity sold by the power supply company will be, the better the psychological feelings it will bring to customers, and the more attractive the electricity sales package will be.

In the context of the opening of the power-selling side, value-added services are diversified, mainly including high-quality power supply information services, high-quality power network trading services, high-quality power supply demand services for users, and high-quality power supply services for high-end users, which can be subdivided into electricity engineering, energy efficiency services (contract energy management, comprehensive energy conservation, contract energy consumption consulting), customer services, high-quality power value-added services, etc. [29]. In the open electricity sales market, electricity sales companies mainly rely on the price of the electricity sales package and its services to improve their competitiveness and attract electricity customers. Among them, value-added services are highly scalable, and power-selling companies can improve customer satisfaction by providing high-quality personalized high-quality value-added services for electricity [30]. To meet the special needs of electric power customers, power-selling companies can provide specific personalized services for power-selling packages. To promote the low-carbon development of the power industry and respond to the national green development requirements, power-selling companies can provide power-selling packages with energy-saving services; To improve the application rate of clean energy, power-selling companies can launch customized electricity selling packages for clean services. In the current era of "Internet plus", to promote the networking of power consumption, power sales companies can provide power sales packages containing new networking services, keeping pace with the times. Furthermore, by integrating several value-added models and integrating resources, they can launch the electricity sales package with the highest value-added comprehensive service model, which fully responds to the personalized needs of electric power customers [31].

From the perspective of electric power customers, the power supply services of power companies can be characterized by five grades: very bad, poor, average, better, and excellent. First of all, according to their subjective experience of electricity use, electric power customers can score the 10 basic and value-added services provided by the power company from 0 to 10, namely, meter reading, fault repair, complaint reporting, inquiry and consultation, electricity engineering, contract energy management, comprehensive energy conservation, contract energy consultation, customer service, and high-quality electricity value-added service. The higher the score, the more satisfied the electric power customers

are with the service. Then, the total satisfaction of electric power customers with power supply service DE is calculated according to the following formula:

$$DE = \sum_{i=1}^{10} \omega_i x_i, \ i = 1, 2, \ldots 10 \tag{4}$$

where $\omega_i$ is the weight of each power supply service, and $x_i$ is the score given by electric power customers for each service; for convenience, let $\omega_1 = \omega_2 = \ldots = \omega_{10} = 0.1$. If $90 < DE \le 100$, then the power supply service is excellent; if $80 < DE \le 90$, then the power supply service is better; if $70 < DE \le 80$, then the power supply service is average; if $60 < DE \le 70$, then the power supply service is poor; if $50 < DE \le 60$, then the power supply service is bad.

### 2.2. Evaluation Index of Power-Selling Company to Users

When selecting users, power-selling companies mainly include the user value and investment ability of users into the main reference indicators.

### 2.2.1. User Value

The user value can be characterized by current value, potential value, and lifetime value [32].

The user value refers to the value that customers can bring to the enterprise, which can be expressed by the monetary contribution that customers bring as benefits to the enterprise. This value is also called customer lifetime value from the perspective of the enterprise, that is, the sum of net profits that enterprises obtain from customers throughout their life cycle. The traditional customer lifetime value consists of current value and potential value [33], as shown in Figure 2.

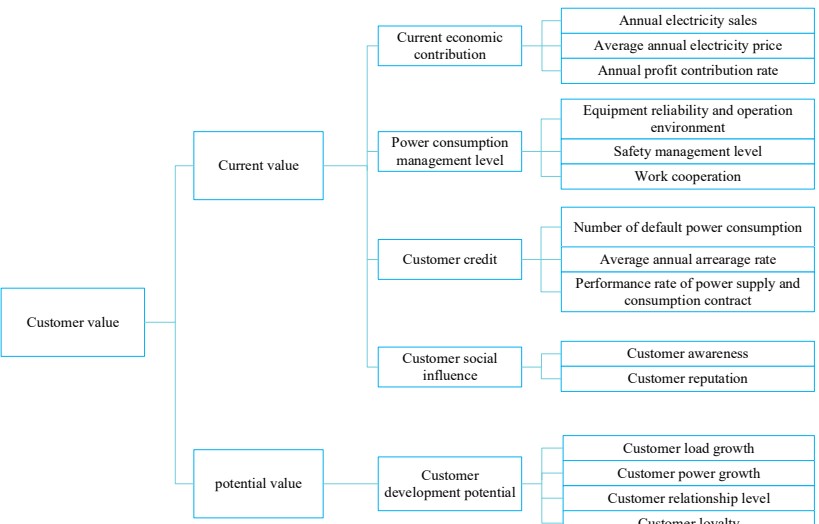

**Figure 2.** Power user value evaluation system.

The authors of [33] proposed a power user value analysis system based on the analytic hierarchy process.

First, establish the fuzzy consistency judgment matrix.

$$R = \begin{pmatrix} r_{11} & r_{12} & \cdots & r_{1n} \\ r_{21} & r_{22} & \cdots & r_{2n} \\ \vdots & \vdots & \cdots & \vdots \\ r_{n1} & r_{n2} & \cdots & r_{nm} \end{pmatrix} \tag{5}$$

Then, use Formula (6) to calculate the weight of each element.

$$w_i = \frac{1}{n^2 - n}(2\sum_{k=1}^{n} r_{jk} - 1), \ i = 1 \sim n \tag{6}$$

Finally, the comprehensive value of electric power customers can be obtained using Formula (7).

$$Z = \sum_{j=1}^{m} x_{ij}w_j \tag{7}$$

Customer lifetime value consists of customer's current value and potential value. The higher the customer's lifetime value, the greater the value of cooperation between the company and the customer will be, and the more benefits can be obtained from it.

2.2.2. User Investment Ability

The user's investment ability can be described from the user's reputation and quality.

As an intangible asset of enterprises, the reputation of enterprise users has become one of the main sources of user competition. The higher the reputation of an enterprise user, the higher the stability, efficiency, profitability, and growth ability of the user's operation will be, and the greater the social responsibility it will bear. The more it can form strong values in society, the greater its social influence will be [34]. Therefore, power-selling companies can make long-term profits by cooperating with reputable enterprise users and can promote their power-selling packages through their social influence to attract more high-quality electric power customers.

Enterprise quality is a comprehensive ability of an enterprise to use human, material, and financial resources to complete its business and production activities. It includes the ability to survive, adaptability, competitiveness, development, and innovation. Therefore, enterprise quality is a complex comprehensive organism, which is not only the unity of production and operation factors and their reasonable organization but also the coordination and unity of internal factors and external conditions of the enterprise [35]. The higher the quality of an enterprise user, the stronger the enterprise user's ability to cope with risks and challenges, and the more stable the development will be. The higher the benefit of the cooperation between the power-selling company and the enterprise user, the more inclined the power-selling company will be to cooperate.

The authors of [36] proposed a method for evaluating investment capacity based on the *TOPSIS* method. First of all, set an ideal value vector $(x_{i1}^*, x_{i2}^*, \ldots, x_{im}^*)$ for several evaluation objects describing the user's investment ability, which represents the optimal investment ability of the power user. Set the actual value of the evaluation object as $(x_{i1}, x_{i2}, \ldots, x_{im})$, and the weighted distance between them is $y_i$:

$$y_i = \sum_{j=1}^{m} w_j f(x_{ij}, x_j^*), \ i = 1, 2, \ldots, n \tag{8}$$

where $w_j$ is the weight coefficient, and $f(x_{ij}, x_j^*)$ is the distance between $x_{ij}$ and $x_j^*$. If $(x_1^+, x_2^+, \ldots, x_m^+)$ is a positive ideal scheme and $(x_1^-, x_2^-, \ldots, x_m^-)$ is a negative ideal scheme, the distance between the evaluation object used to describe the investment ability and the positive ideal points and the distance between the evaluation object and the negative ideal points are respectively shown as follows:

$$y_i^+ = \sqrt{\sum_{j=1}^{m} w_j(x_{ij} - x_j^+)^2}, \ i = 1, 2, \ldots, n \tag{9}$$

$$y_i^- = \sqrt{\sum_{j=1}^{m} w_j(x_{ij} - x_j^-)^2}, \; i = 1, 2, \ldots, n \tag{10}$$

User investment ability can be expressed as $C_i$:

$$C_i = \frac{y_i^-}{y_i^+ + y_i^-} \tag{11}$$

The larger $C_i$ is, the stronger the power user's investment ability is and the more favorably it can be obtained from the power-selling company.

## 3. Matching Method between Power User Demand and Power-Selling Package of Power-Selling Company

### 3.1. Overview of Two-Sided Matching Problem (TSMDM)

The purpose of a two-sided matching problem is to find the best matching method between objects on both sides according to the preference information or evaluation results provided by matching objects, so as to maximize the interests of both sides. In this paper, the power user was set as a matrix $P = \{P_1, P_2, \ldots, P_n\}$, where $P_i$ represents the $i$-th user in the power user set, $i \in \{1, 2, \ldots, n\} = I$. The electricity sales package was set as a collection $Q = \{Q_1, Q_2, \ldots, Q_m\}$ ($n \le m$), where $Q_j$ refers to the $j$-th electricity sales package in the electricity sales package set, $j \in \{1, 2, \ldots, m\} = J$. Two-sided matching is a one-to-one mapping $\mu$ from set $P$ to set $Q$, where $\mu(P_i) = Q_j$ means that the power user $P_i$ matches the electricity sales package $Q_j$, and vice versa. It should be noted that there will be $m$–$n$ packages that will not be selected.

The existing definitions are as follows: $u_{ij}^P$ indicates the user $P_i$'s satisfaction with the package $Q_j$, and $u_{ij}^Q$ indicates the adaptation degree of the package $Q_j$ to the user $P_i$. In the following two situations, the two-sided match is unstable; otherwise, it is a stable two-sided match:

1.  $\exists P_i, P_l \in P, Q_j, Q_k \in Q, \mu(P_i) = Q_k, \mu(P_l) = Q_j$, making $u_{ij}^P > u_{ik}^P$ and $u_{ij}^Q > u_{lj}^Q$;
2.  $\exists P_i \in P, Q_j, Q_k \in Q, \mu(P_i) = Q_k, \mu(Q_j) = Q_j$, making $u_{ij}^P > u_{ik}^P$.

Next, we introduce a binary variable $x_{ij}$ to indicate whether $P_i$ and $Q_j$ is matched. The conditions for stable two-sided matching can be defined as follows:

$$x_{ij} + \sum_{u_{ih}^P > u_{ij}^P} x_{ih} + \sum_{u_{kj}^Q > u_{ij}^Q} x_{kj} \ge 1, \; i \in I, \; j \in J \tag{12}$$

$$x_{ij} = \begin{cases} 1, & \mu(P_i) = Q_j \\ 0, & \mu(P_i) \ne Q_j \end{cases} \tag{13}$$

### 3.2. Incomplete Fuzzy Preference Relationship

For convenience, the assumption $X = \{x_1, x_2, \ldots, x_p\}$ is a set of fixed options of the decision maker, where $x_i$ represents the fuzzy preference degree of the $i$-th element in the decision-maker set $X$. We use a matrix $A = (a_{ij})_{p \times p}$ to describe it. $a_{ij} \in (0.5, 1)$ indicates that, for the decision maker, $x_i$ is better than $x_j$, $a_{ij} \in (0, 0.5)$ indicates that $x_i$ is not preferred to $x_j$, and $a_{ij} = 0.5$ indicates that $x_i$ and $x_j$ are equally preferred. In order to make the results more rigorous, we stipulate $a_{ij} + a_{ji} = 1$, $i, j = 1, 2, \ldots, p$.

In fuzzy preference matrix $A$, if some elements are unknown, then $A$ is called an incomplete fuzzy preference relation matrix. If at least one element other than diagonal is known in each row and column, then $A$ is called an acceptable incomplete fuzzy preference matrix; otherwise, it is unacceptable.

### 3.3. Subjective Satisfaction

Generally speaking, a power user has different preferences for different power sales packages, and, for a certain power sales package, the power company has different preferences for different electric power customers. Therefore, we can rank the satisfaction of electricity users and electricity sales packages.

We assume that $U^P = (u^p_{ij})_{n \times m}$ can be used to express the subjective satisfaction matrix of the electricity sales package given by the electric power customers according to their own demand for electricity, where the element $u^p_{ij}$ represents the subjective satisfaction of the power user $P_i$ with the electricity sales package $Q_j$, which reflects $P_i$'s preference for $Q_j$. The greater the value is, the higher the degree of preference is. In the same way, $U^Q = (u^Q_{ij})_{n \times m}$ represents the subjective satisfaction matrix of electric power customers given by the power sales package according to their preferences for electric power customers, and the element $u^Q_{ij}$ represents the subjective satisfaction of the power sales package $Q_j$ to power user $P_i$.

We assume that $R^i = (r^i_{jl})_{m \times m}$ is used to represent the incomplete fuzzy preference relationship of power user $P_i$ for $m$ kinds of electricity sales packages, where $r^i_{jl}$ represents the result of $P_i$ comparing electricity sales package $Q_j$ with the electricity sales package $Q_l$; $0 < r^i_{jl} < 0.5$ means that the preference of power user $P_i$ for the package $Q_j$ is less than that for the package $Q_l$, $0.5 < r^i_{jl} < 1$ means that the preference of power user $P_i$ for the package $Q_j$ is higher than that for the package $Q_l$, and $r^i_{jl} = 0.5$ means that the preference of electric power customers $P_i$ for the package $Q_j$ is the same as that for the package $Q_l$. In addition, if electric power customers cannot compare the two packages, data will be missing, as indicated by $r^i_{jl} = \varphi$.

In order to obtain the priority weight vector of each power user for all electricity sales packages, we need to introduce an indicator matrix $\Delta = (\delta_{ij})_{m \times m}$, where

$$\delta_{ij} = \begin{cases} 0, & a_{ij} = \varphi, \\ 1, & a_{ij} \neq \varphi, \end{cases} , \ i,j = 1,2,\ldots,m \tag{14}$$

The priority weight vector of the power user $P_i$ can be obtained using the *LLSM* method proposed in [29]. The specific steps are as follows:

First, use the indicator matrix to obtain the matrices $D$ and $Y$.

$$D = \begin{pmatrix} \sum\limits_{j=2}^{m} \delta_{1j} & -\delta_{12} & \cdots & -\delta_{1,m-1} \\ -\delta_{21} & \sum\limits_{j=2 j \neq 2}^{m} \delta_{2j} & \cdots & -\delta_{2,m-1} \\ \vdots & \vdots & \cdots & \vdots \\ -\delta_{m-1,1} & -\delta_{m-1,2} & \cdots & \sum\limits_{\substack{j=2 \\ j \neq m-1}}^{m} \delta_{m-1,j} \end{pmatrix}, Y = \begin{pmatrix} \sum\limits_{j=1}^{m} \delta_{1j}(\ln a_{1j} - \ln a_{j1}) \\ \sum\limits_{j=1}^{m} \delta_{2j}(\ln a_{2j} - \ln a_{j2}) \\ \cdots \\ \sum\limits_{j=1}^{m} \delta_{m-1,j}(\ln a_{m-1,j} - \ln a_{j,m-1}) \end{pmatrix} \tag{15}$$

Next, the matrix $D$ and $Y$ are substituted into the formula $W = (W_1, W_2, \ldots, W_{m-1})^T = D^{-1}Y$ to obtain the $W$ vector, and then the vector $W$ is substituted into formula (16) to obtain the $P_i$ priority weight vector of electric power customers $\omega^{P_i} = (\omega_1^{P_i}, \omega_2^{P_i}, \ldots, \omega_{m-1}^{P_i})^T$.

$$
w = \begin{cases} \dfrac{\exp(W_i)}{\sum\limits_{j=1}^{m-1} \exp(W_j) + 1}, & i = 1, 2, \ldots, m-1 \\[4mm] \dfrac{1}{\sum\limits_{j=1}^{m-1} \exp(W_j) + 1}, & i = m \end{cases}
\tag{16}
$$

According to [37,38], if $R^{i'}$ is an unacceptable incomplete fuzzy preference relation matrix, the priority weight vector can be obtained through the following steps:

1.  Remove the rows and columns with only one known element (assuming that the first row and the first column are $1 < l < m$) to obtain a new acceptable incomplete fuzzy preference matrix $R^{i'}$;
2.  Use the above method to obtain an incomplete priority weight vector $\omega^{P_i'} = (\omega_1^{P_i}, \omega_2^{P_i}, \ldots, \omega_{m-1}^{P_i})^T$;
3.  Insert $M$ in the line next to line $l-1$ of the vector $\omega^{P_i'}$, or $M$ in the line above line $l+1$. $M$ shows that the decision makers have no clear preference for the first type of electricity sales package.

Lastly, the subjective satisfaction of electric power customers $P_i'$ with the electricity sales package $Q_j$ can be expressed by $u_{ij}^P$.

$$
u_{ij}^P = \begin{cases} \dfrac{\omega_j^{P_i} - \min\limits_{Q_j \in \Delta^{P_i}}\left\{\omega_j^{P_i}\right\}}{\max\limits_{j \in J}\left\{\omega_j^{P_i}\right\} - \min\limits_{Q_j \in \Delta^{P_i}}\left\{\omega_j^{P_i}\right\}}, & Q_j \in \Delta^{P_i} \\[4mm] -M, & Q_j \notin \Delta^{P_i} \end{cases}, \ i \in I
\tag{17}
$$

Similarly, we set $T^j = (t_{ik}^j)_{n \times n}$ to represent the incomplete fuzzy preference relationship of the electricity sales package $Q_j$ for $n$ electric power customers.

In the same way, it can be concluded that the subjective satisfaction degree of the electricity sales package $Q_j$ with the power user $P_i$ can be expressed by $u_{ij}^Q$.

$$
u_{ij}^Q = \begin{cases} \dfrac{\omega_i^{Q_j} - \min\limits_{P_i \in \Delta^{Q_j}}\left\{\omega_i^{Q_j}\right\}}{\max\limits_{i \in I}\left\{\omega_i^{Q_j}\right\} - \min\limits_{P_i \in \Delta^{Q_j}}\left\{\omega_i^{Q_j}\right\}}, & P_i \in \Delta^{Q_j} \\[4mm] -M, & P_i \notin \Delta^{Q_j} \end{cases}, \ j \in J
\tag{18}
$$

where $\Delta^{P_i} = \left\{Q_j \middle| \omega_j^{P_i} \neq -M, j \in J\right\}$ is a set containing user $P_i$'s effective preference information, and $\Delta^{Q_j} = \left\{P_i \middle| \omega_i^{Q_j} \neq -M, i \in I\right\}$ is a set containing the effective preference information of the package $Q_j$.

### 3.4. Two-Sided Matching Decision Based on Disappointment Theory

The theory of disappointment was first put forward by Bell [39]. It is argued that disappointment is a psychological reaction of decision makers by comparing actual results with expected results. The two-way choice between electric power customers and power sales packages is a product that satisfies both sides of satisfaction. It is a psychological evaluation of the currently selected objects by both sides, which is related to the psychological perception of disappointment and elation. Disappointment is the sense of dissatisfaction

when the actual result does not meet the expected standard of the decision maker, while elation is the satisfaction generated when the actual result exceeds the expected standard of the decision maker. In addition, disappointment usually has more influence than elation for the same difference between the actual outcome and the expected outcome, which is known as disappointment aversion. Bell [39] implicitly pointed out that the function to calculate an individual's real utility should be a combination of the subjective utility function and disappointment–elation function.

Soon afterward, Loomes and Sugden [40] also argued that "disappointment" and "elation" are key components for making rational choices. Assuming that the $j$-th state of an action $A_i$ occurs with probability $p_j$, $0 < p_j < 1$, $\sum_{j=1}^{n} p_j = 1$, Loomes and Sugden [40] showed that the modified expected utility of action $A_i$ can be calculated by

$$E_i = \sum_{j=1}^{n} p_j[c_{ij} + D(c_{ij} - \overline{c_i})] \tag{19}$$

where $c_{ij}$ is the basic utility of consequence, $x_{ij}\overline{c_i} = \sum_{j=1}^{n} p_i c_{ij}$ denotes the expected basic utility and function, and $D(\cdot)$ is a function that is used to calculate disappointment and elation.

However, Delqui'e and Cillo [41] showed that an individual's disappointment with the outcome is related to the result that the individual did not achieve any possible outcome, but not to the expected utility. Because it is difficult to choose an indicator to play the role of prior expectations, any outcome can become the expected outcome to a certain extent according to its probability. Therefore, any result may trigger disappointment or excitement values. On the basis of this idea, Delqui'e and Cillo [41] believe that, when comparing result $x_i$ with another result that is better than $x_i$, individuals will feel disappointed, whereas, when comparing result $x_i$ with a result that is worse than $x_i$, individuals will feel happy instead. According to the subjective utilities, all results can be ranked in descending order as $x_1 \geq x_2 \geq \ldots \geq x_n$; the adjusted utility is defined as

$$u(x_i) = v(x_i) - \left(\sum_{k=1}^{i} p_k D(v(x_k) - v(x_i))\right) + \left(\sum_{k=i}^{n} p_k E(v(x_i) - v(x_k))\right) \tag{20}$$

where $v(x_i)$ is the subjective utility of the outcome $x_i$, $D(\cdot)$ is a disappointment function denoting one's sensitivity to disappointment, and $E(\cdot)$ is an elation function which captures one's sensitivity to elation.

Similarly, disappointment and elation will also appear for decision-making problems with fuzzy preference relations. Next, we discuss how to calculate the adjusted utility on the basis of incomplete fuzzy preference relationships.

Assume that the priority weight vector value $Q = \{Q_1, Q_2, \ldots, Q_j, \ldots, Q_m\}$ of power user $P_i$ for the electricity sales package is ranked from low to high: $u_{i1}^P < u_{i2}^P < \ldots < u_{ij-1}^P < u_{ij}^P < u_{ij+1}^P < \ldots < u_{im}^P$. If $P_i$ and $Q_j$ match, at this time, the satisfaction of power user $P_i$ is not only related to the electricity sales package $Q_j$, but also related to other electricity sales packages. On the one hand, because the packages in the collection $\{Q_1, Q_2, \ldots, Q_{j-1}\}$ are inferior to $Q_j$, power user $P_i$ will feel happy because they do not match them. On the other hand, because the packages in the collection $\{Q_{j+1}, Q_{j+2}, \ldots, Q_m\}$ are better than $Q_j$, power user $P_i$ will be disappointed because they do not match them.

In the same way, the same is true for the electricity sales package $Q_j$.

Therefore, to evaluate the satisfaction of electric power customers and electricity sales packages with the matching results, we need to consider the disappointment–elation perception of both sides, so that we can more accurately describe the satisfaction degree of electric power customers and electricity sales packages.

Because there is a lack of subjective evaluation elements in the subjective satisfaction matrix of electric power customers $U^P = (u_{ij}^p)_{n \times m}$ and the subjective satisfaction matrix of electricity sales packages $U^Q = (u_{ij}^Q)_{n \times m}$, we build a collection:

$$\Theta^{P_i} = \left\{ Q_j \middle| u_{ij}^P \neq -M \& u_{ij}^Q \neq -M, j \in J \right\}, \, i \in I \tag{21}$$

$$\Theta^{Q_j} = \left\{ P_i \middle| u_{ij}^P \neq -M \& u_{ij}^Q \neq -M, i \in I \right\}, \, j \in I \tag{22}$$

Set (21) refers to the set of electricity sales packages that can match each $P_i$ in the electricity sales package set $Q$. Set (22) represents the set of electric power customers that can match each $Q_j$ in the power user set $P$ for the power sales package.

If the power user $P_i$ matches the $P_i$ electricity sales package $Q_j$, $\overline{u_{ij}^P}$ means that the correction of subjective satisfaction degree after adding the disappointment–elation feeling of the power user, according to the definition, $\overline{u_{ij}^P}$ can be expressed as

$$\overline{u_{ij}^P} = \begin{cases} u_{ij}^P - d_{ij}^P + e_{ij}^P & Q_j \in \Theta'^{P_i} \\ u_{ij}^P & Q_j \notin \Theta^{P_i} \end{cases}, \, i \in I \tag{23}$$

In formula (23), $d_{ij}^P$ represents the user $P_i$'s disappointment value, and $e_{ij}^P$ represents the user $P_i$'s elation value.

For the matching object $Q_j \in \Theta^{P_i}$, if there is a certain electricity sales package $Q_l \in \Theta^{P_i}$, leading to $u_{ij}^P < u_{il}^P$, then the power user $P_i$ will be disappointed when matching with the electricity sales package $Q_j$ instead of matching $Q_l$. Set the collection $\Delta_{ij}^{DP} = \left\{ Q_l \middle| u_{ij}^P < u_{il}^P \& Q_l \in \Theta^{P_i} \right\}$ as a collection of objects that will cause $P_i$ disappointment after matching with $Q_j$. At this time, the user $P_i$'s disappointment value can be calculated using the following formula:

$$d_{ij}^P = prob(P_i, Q_l) \sum_{Q_l \in \Delta_{ij}^{DP}} D_i(u_{il}^P - u_{ij}^P), \, \forall Q_l \in \Delta_{ij}^{DP}, \, Q_j \in \Theta^{P_i}, \, i \in I \tag{24}$$

In the same way, $Q_l \in \Theta^{P_i}$ can also exist, leading to $u_{ij}^P > u_{il}^P$, such that $P_i$ will feel happy when matching with $Q_j$ instead of $Q_l$. Set the collection $\Delta_{ij}^{EP} = \left\{ Q_l \middle| u_{ij}^P > u_{il}^P \& Q_l \in \Theta^{P_i} \right\}$ as the collection of objects for which $P_i$ will produce a sense of elation after matching with $Q_j$. At this time, the user $P_i$'s happiness value can be calculated using the following formula:

$$e_{ij}^P = prob(P_i, Q_l) \sum_{Q_l \in \Delta_{ij}^{EP}} E_i(u_{ij}^P - u_{il}^P), \, \forall Q_l \in \Delta_{ij}^{EP}, \, Q_j \in \Theta^{P_i}, \, i \in I \tag{25}$$

In formulas (24) and (25), the $prob(P_i, Q_l)$ reciprocal of the number of electricity sales packages that disappoint power user $P_i$ and the reciprocal of the number of electricity sales packages that delight power user $P_i$ are respectively expressed as follows:

$$prob(P_i, Q_l) = \begin{cases} \frac{1}{\left| \Delta_{ij}^{DP} \right|} & Q_j \in \Delta'^{DP}_{ij} \\ 0 & Q_j \notin \Delta_{ij}^{DP} \end{cases}, \, i \in I, \, j \in J \tag{26}$$

$$prob(P_i, Q_l) = \begin{cases} \frac{1}{\left| \Delta_{ij}^{EP} \right|} & Q_j \in \Delta'^{EP}_{ij} \\ 0 & Q_j \notin \Delta_{ij}^{EP} \end{cases}, \, i \in I, \, j \in J \tag{27}$$

In the same way, if the electricity sales package $Q_j$ matches the power user $P_i$, $\overline{u_{ij}^Q}$ represents the correction of subjective satisfaction after adding electricity sales package $Q_j$'s disappointment–elation feelings, which can be expressed as

$$\overline{u_{ij}^Q} = \begin{cases} u_{ij}^Q - d_{ij}^Q + e_{ij}^Q \\ u_{ij}^Q \end{cases} \tag{28}$$

$$d_{ij}^Q = prob(Q_j, P_k) \sum_{P_k \in \Delta_{ij}^{DQ}} D_i(u_{kj}^Q - u_{ij}^Q), \ \forall P_k \in \Delta_{ij}^{DQ}, \ P_i \in \Theta^{Q_j}, \ j \in J \tag{29}$$

$$e_{ij}^Q = prob(Q_j, P_k) \sum_{p_k \in \Delta_{ij}^{EQ}} E_i(u_{ij}^Q - u_{kj}^Q), \ \forall P_k \in \Delta_{ij}^{EQ}, \ P_i \in \Theta^{Q_j}, \ j \in J \tag{30}$$

$$prob(Q_j, P_k) = \begin{cases} \dfrac{1}{\left|\Delta_{ij}^{DQ}\right|}, & Q_j \in \Delta_{ij}^{DQ} \\ 0 & Q_j \notin \Delta_{ij}^{DQ} \end{cases}, \ i \in I, \ j \in J \tag{31}$$

$$prob(Q_j, P_k) = \begin{cases} \dfrac{1}{\left|\Delta_{ij}^{EQ}\right|}, & Q_j \in \Delta_{ij}^{EQ} \\ 0 & Q_j \notin \Delta_{ij}^{EQ} \end{cases}, \ i \in I, \ j \in J \tag{32}$$

The specific expressions of disappointment function $D(\cdot)$ and elation function $E(\cdot)$ are as follows:

$$D(x) = 1 - \alpha^x, \ x \geq 0 \tag{33}$$

$$E(x) = \gamma(1 - \beta^x), \ x \geq 0 \tag{34}$$

In formulas (33) and (34), $\alpha$ and $\beta$ are the disappointment and elation parameters, where $\gamma$ indicates the degree of influence of the feeling of elation on the evaluation results, $0 < \beta < 10 < \alpha < 1, 0 < \gamma < 1$. The smaller $\alpha$ is, the greater the sensitivity of electric power customers to the disappointment of the electricity sales package is, and the less likely they are to choose the electricity sales package. The larger $\beta$ is, the greater the sensitivity of electric power customers to the elation of the electricity sales package is, and the more likely they are to choose the electricity sales package.

To sum up, the satisfaction evaluation matrix of electric power customers after adding the disappointment–elation perception is

$$\overline{U^P} = \left(\overline{u_{ij}^P}\right)_{n \times m} = \begin{pmatrix} \overline{u_{11}^P} & \overline{u_{12}^P} & \cdots & \overline{u_{1m}^P} \\ \overline{u_{21}^P} & \overline{u_{22}^P} & \cdots & \overline{u_{2m}^P} \\ \vdots & \vdots & \cdots & \vdots \\ \overline{u_{n1}^P} & \overline{u_{n2}^P} & \cdots & \overline{u_{nm}^P} \end{pmatrix} \tag{35}$$

The satisfaction evaluation matrix of the electricity sales package is

$$\overline{U^Q} = \left(\overline{u_{ij}^Q}\right)_{n \times m} = \begin{pmatrix} \overline{u_{11}^Q} & \overline{u_{12}^Q} & \cdots & \overline{u_{1m}^Q} \\ \overline{u_{21}^Q} & \overline{u_{22}^Q} & \cdots & \overline{u_{2m}^Q} \\ \vdots & \vdots & \cdots & \vdots \\ \overline{u_{n1}^Q} & \overline{u_{n2}^Q} & \cdots & \overline{u_{nm}^Q} \end{pmatrix} \tag{36}$$

### 3.5. Multi-Objective Optimization Model

Next, after obtaining the satisfaction matrix of power user set $P$ and the satisfaction matrix of power sales package set $Q$, a two-objective optimization model can be constructed, and the optimal stable matching of both parties can be obtained by solving the model.

Set $x_{ij}$ as a binary decision variable. When the user $P_i$ matches the package $Q_j$, $x_{ij} = 1$; otherwise, $x_{ij} = 0$.

With the maximum satisfaction of electric power customers and electricity sales package as the optimization objective, a stable two-sided matching multi-objective optimization model can be established:

$$\max Z_1 = \sum_{i=1}^{n} \sum_{j=1}^{m} \overline{u_{ij}^P} x_{ij} \tag{37}$$

$$\max Z_2 = \sum_{j=1}^{m} \sum_{i=1}^{n} \overline{u_{ij}^Q} x_{ij} \tag{38}$$

$$\sum_{j=1}^{m} x_{ij} \leq 1, \, i \in I \tag{39}$$

$$\sum_{i=1}^{n} x_{ij} \leq 1, \, j \in J \tag{40}$$

$$x_{ij} + \sum_{\overline{u_{ih}^P} > \overline{u_{ij}^P}} x_{ih} + \sum_{\overline{u_{kj}^Q} > \overline{u_{ij}^Q}} x_{kj} \geq 1, \, \forall (P_i, Q_j) \in \Omega \tag{41}$$

$$x_{ij} \in \{0, 1\} \, i \in I, \, j \in J \tag{42}$$

In the above model, formulas (37) and (38) are objective functions, which describe the maximization of satisfaction between electric power customers and electricity sales packages. Equations (39) and (40) are inequality constraints, which respectively ensure that each object in user $P$ can only match one package in package $Q$ at most, and that one power-selling package in package set $Q$ can only match one user in user set $P$. Equation (41) is a stable two-sided matching constraint, including two situations: (1) power user $P_i$ and electricity sales package $Q_j$ match each other; (2) if power user $P_i$ does not match the electricity sales package $Q_j$, then power user $P_i$ will match electricity sales package $Q_j$ with a higher degree of preference or electricity sales package $Q_j$ will match power user $P_i$ with a higher degree of preference. In Equation (41), the meaning of $\overline{u_{ih}^P} > \overline{u_{ij}^P}$ is that, in the electricity user set P, the preference of the $i$-th power user for the $h$-th electricity sales package is higher than that for the $j$-th electricity sales package. Similarly, the meaning of $\overline{u_{ih}^P} > \overline{u_{ij}^P}$ is that, in the electricity sales package set $Q$, the preference of the $j$-th electricity sales package for the $k$-th electricity user is higher than that for the $i$-th electricity user.

In order to facilitate the solution, the weight coefficient $\omega$ is introduced to transform the two-objective optimization model into a single-objective optimization model:

$$\max Z = \omega \sum_{i=1}^{n} \sum_{j=1}^{m} \overline{u_{ij}^P} x_{ij} + (1 - \omega) \sum_{j=1}^{m} \sum_{i=1}^{n} \overline{u_{ij}^Q} x_{ij} \tag{43}$$

$$\sum_{j=1}^{m} x_{ij} \leq 1, \, i \in I \tag{44}$$

$$\sum_{i=1}^{n} x_{ij} \leq 1, \, j \in J \tag{45}$$

$$x_{ij} + \sum_{\overline{u_{ih}^P} > \overline{u_{ij}^P}} x_{ih} + \sum_{\overline{u_{kj}^Q} > \overline{u_{ij}^Q}} x_{kj} \geq 1, \, \forall (P_i, Q_j) \in \Omega \tag{46}$$

$$x_{ij} \in \{0,1\}, \ i \in I, \ j \in J \tag{47}$$

In model (43), Z represents the total satisfaction of the matching parties, and $\omega$ is a weight coefficient, where $0 < \omega < 0.5$ indicates that the feeling of electric power customers is more important than the adaptation degree of the electricity sales package, $0.5 < \omega < 1$ indicates that the adaptability of the electricity sales package is more important than the satisfaction of the electric power customers, and $\omega = 0.5$ indicating that both parties are equally important. In this paper, we take $\omega = 0.5$ for analysis.

According to the two-sided matching theory, after adding the disappointment–elation perception effect of electric power customers and electricity sales packages, the stable two-sided matching process between electric power customers and electricity sales package is as shown in Figure 3.

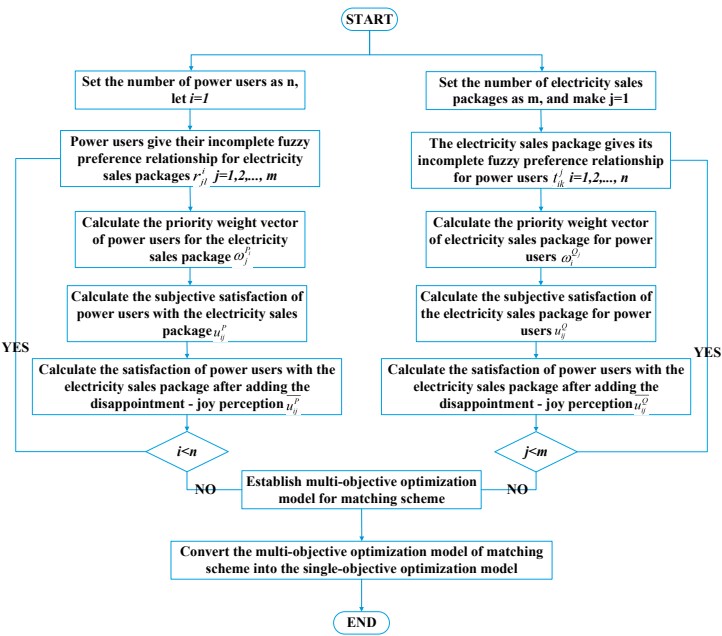

**Figure 3.** Stable two-sided matching process between electric power customers and electricity sales package.

## 4. Actual Case Analysis

### 4.1. Case Background

As shown in Figure 4, there is an industrial park in a certain area of Zhejiang, which includes five electric power customers $P = \{P_1, P_2, P_3, P_4, P_5\}$ with six electricity sales packages $Q = \{Q_1, Q_2, Q_3, Q_4, Q_5, Q_6\}$ to choose from. We need to match them one by one according to their respective preferences, so as to maximize their respective satisfaction and overall satisfaction, and finally give a stable matching model. For electric power customers, the main factors to be considered in the evaluation of electricity sales package are power quality and power supply service. For the electricity sales package, the main considerations for electric power customers are user value and user investment ability. Let the incomplete fuzzy preference relationship between each power user and each electricity sales package be expressed as $R^i$ and $T^j$. Let the priority weight vector be expressed as $w^{P_i}$ and $w^{Q_j}$, and let subjective satisfaction be expressed as $U^P$ and $U^Q$. The sensitivity parameters of both parties to disappointment are $\alpha_i^P = \alpha_j^Q = 0.8$, the degree of disgust to disappointment is $\gamma_i^P = \gamma_j^Q = 0.5$, the sensitivity parameter of elation is $\beta_i^P = \beta_j^Q = 0.8$, and the disappointment value is $D^P$ and $D^Q$; the weight when the double-objective optimization model is converted to the single-objective optimization model is $\omega = 0.5$.

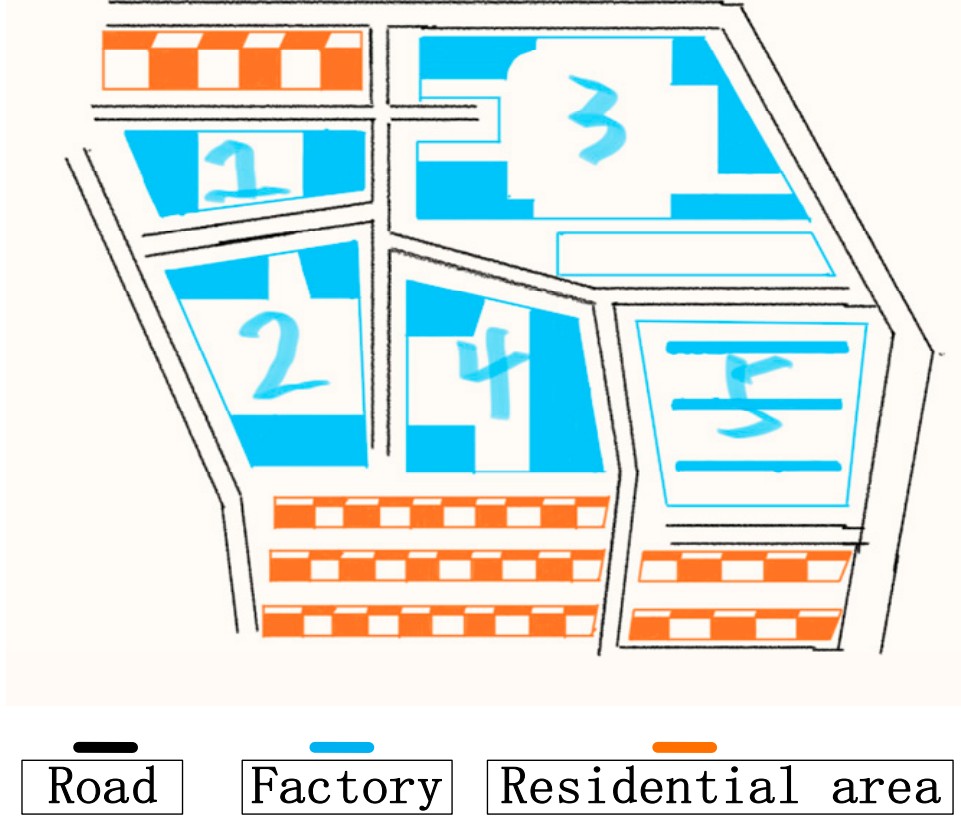

Road    Factory    Residential area

**Figure 4.** Schematic diagram of industrial park.

In Figure 4, the numbers 1 to 5 represent the numbers of five factories, respectively.

*4.2. Case Analysis*

4.2.1. Incomplete Fuzzy Preference Relationship

Five electric power customers are required to provide their preference information for six electricity sales packages through pairwise comparisons by considering some criteria, including clean energy ratio, power quality, and power supply services. The incomplete fuzzy preference relationship $P_i$ provided by each power user is expressed by $R^i(i = 1, 2, \ldots, 5)$. At the same time, six electricity sales packages are also required to express their preference information for electric power customers through pairwise comparisons considering their value, investment capability, etc. The incomplete fuzzy preference relationship $Q_j$ of each power sales package is recorded as $T^j(j = 1, 2, \ldots 6)$, as expressed below.

$$R^1 = \begin{pmatrix} 0.5 & \varphi & \varphi & \varphi & \varphi & \varphi \\ \varphi & 0.5 & 0.35 & 0.65 & \varphi & 0.75 \\ \varphi & 0.65 & 0.5 & \varphi & 0.55 & 0.55 \\ \varphi & 0.35 & \varphi & 0.5 & 0.75 & \varphi \\ \varphi & \varphi & 0.45 & 0.25 & 0.5 & 0.65 \\ \varphi & 0.25 & 0.45 & \varphi & 0.35 & 0.5 \end{pmatrix} \tag{48}$$

$$R^2 = \begin{pmatrix} 0.5 & 0.25 & \varphi & 0.55 & \varphi & 0.35 \\ 0.75 & 0.5 & 0.15 & 0.65 & \varphi & 0.15 \\ \varphi & 0.85 & 0.5 & \varphi & \varphi & 0.75 \\ 0.45 & 0.35 & \varphi & 0.5 & \varphi & \varphi \\ \varphi & \varphi & \varphi & \varphi & 0.5 & \varphi \\ 0.65 & 0.85 & 0.25 & \varphi & \varphi & 0.5 \end{pmatrix} \tag{49}$$

$$R^3 = \begin{pmatrix} 0.5 & 0.35 & \varphi & 0.25 & 0.75 & 0.25 \\ 0.65 & 0.5 & \varphi & 0.45 & \varphi & 0.35 \\ \varphi & \varphi & 0.5 & \varphi & \varphi & \varphi \\ 0.75 & 0.55 & \varphi & 0.5 & 0.35 & 0.75 \\ 0.25 & \varphi & \varphi & 0.65 & 0.5 & 0.65 \\ 0.75 & 0.65 & \varphi & 0.25 & 0.35 & 0.5 \end{pmatrix} \tag{50}$$

$$R^4 = \begin{pmatrix} 0.5 & \varphi & 0.65 & 0.45 & 0.55 & 0.65 \\ \varphi & 0.5 & \varphi & \varphi & \varphi & \varphi \\ 0.35 & \varphi & 0.5 & 0.75 & 0.25 & 0.55 \\ 0.55 & \varphi & 0.25 & 0.5 & 0.15 & \varphi \\ 0.45 & \varphi & 0.75 & 0.85 & 0.5 & 0.15 \\ 0.35 & \varphi & 0.45 & \varphi & 0.85 & 0.5 \end{pmatrix} \tag{51}$$

$$R^5 = \begin{pmatrix} 0.5 & 0.65 & 0.75 & \varphi & 0.55 & 0.25 \\ 0.35 & 0.5 & \varphi & 0.65 & 0.25 & 0.15 \\ 0.25 & \varphi & 0.5 & 0.35 & 0.55 & 0.65 \\ \varphi & 0.35 & 0.65 & 0.5 & 0.35 & \varphi \\ 0.45 & 0.75 & 0.45 & 0.65 & 0.5 & 0.45 \\ 0.75 & 0.85 & 0.35 & \varphi & 0.55 & 0.5 \end{pmatrix} \tag{52}$$

$$T^1 = \begin{pmatrix} 0.5 & 0.65 & \varphi & 0.55 & 0.75 \\ 0.35 & 0.5 & 0.25 & 0.35 & 0.65 \\ \varphi & 0.75 & 0.5 & \varphi & 0.35 \\ 0.45 & 0.65 & \varphi & 0.5 & 0.65 \\ 0.25 & 0.35 & 0.65 & 0.35 & 0.5 \end{pmatrix} \tag{53}$$

$$T^2 = \begin{pmatrix} 0.5 & \varphi & \varphi & \varphi & \varphi \\ \varphi & 0.5 & 0.05 & 0.45 & 0.55 \\ \varphi & 0.95 & 0.5 & 0.35 & \varphi \\ \varphi & 0.55 & 0.65 & 0.5 & 0.75 \\ \varphi & 0.45 & \varphi & 0.25 & 0.5 \end{pmatrix} \tag{54}$$

$$T^3 = \begin{pmatrix} 0.5 & 0.25 & \varphi & 0.65 & \varphi \\ 0.75 & 0.5 & 0.65 & \varphi & \varphi \\ \varphi & 0.35 & 0.5 & 0.75 & \varphi \\ 0.35 & \varphi & 0.25 & 0.5 & \varphi \\ \varphi & \varphi & \varphi & \varphi & 0.5 \end{pmatrix} \tag{55}$$

$$T^4 = \begin{pmatrix} 0.5 & \varphi & 0.35 & 0.55 & 0.15 \\ \varphi & 0.5 & \varphi & \varphi & \varphi \\ 0.65 & \varphi & 0.5 & 0.25 & 0.75 \\ 0.45 & \varphi & 0.75 & 0.5 & 0.15 \\ 0.85 & \varphi & 0.25 & 0.85 & 0.5 \end{pmatrix} \tag{56}$$

$$T^5 = \begin{pmatrix} 0.5 & 0.25 & 0.45 & 0.65 & 0.15 \\ 0.75 & 0.5 & 0.15 & 0.55 & 0.75 \\ 0.55 & 0.85 & 0.5 & 0.05 & \varphi \\ 0.35 & 0.45 & 0.95 & 0.5 & 0.25 \\ 0.85 & 0.25 & \varphi & 0.75 & 0.5 \end{pmatrix} \tag{57}$$

$$T^6 = \begin{pmatrix} 0.5 & \varphi & 0.35 & 0.55 & 0.15 \\ \varphi & 0.5 & \varphi & \varphi & \varphi \\ 0.65 & \varphi & 0.5 & 0.25 & 0.75 \\ 0.45 & \varphi & 0.75 & 0.5 & 0.15 \\ 0.85 & \varphi & 0.25 & 0.85 & 0.5 \end{pmatrix} \tag{58}$$

4.2.2. Calculation of Subjective Satisfaction

On the basis of the incomplete fuzzy preference relationship between electric power customers and electricity sales package, the priority vector of all electric power customers and electricity sales package can be calculated, taking $w^{Q_1}$ and $w^{P_1}$ as an example:

$$w^{Q_1} = (0.3589, 0.1354, 0.1456, 0.2589, 0.1152),$$
$$w^{P_1} = (-M, 0.3052, 0.2189, 0.2456, 0.1386, 0.1210).$$

On the basis of the priority weight vectors $w^{P_1}$ and $w^{Q_1}$, the subjective satisfaction of electric power customers with the electricity sales package $u_{ij}^P$ and the subjective satisfaction $u_{ij}^Q$ of electric power customers with the electricity sales package can be calculated, yielding $U^P = (u_{ij}^p)_{5\times 6}$, $U^Q = (u_{ij}^Q)_{5\times 6}$. The following matrix can be obtained:

$$U^P = \begin{pmatrix} -M & 1 & 0.6468 & 0.7692 & 0.1818 & 0 \\ 0.0614 & 0.1335 & 1 & 0 & -M & 0.2628 \\ 0.0838 & 0.3519 & -M & 1 & 0.2210 & 0 \\ 1 & -M & 0.5410 & 0 & 0.5904 & 0.7575 \\ 1 & 0 & 0.3656 & 0.0232 & 0.4863 & 0.9572 \end{pmatrix} \tag{59}$$

$$U^Q = \begin{pmatrix} 1 & -M & 0.2825 & 0 & 0 & 0.6153 \\ 0.1244 & 0.1320 & 1 & -M & 0.9421 & 0.3230 \\ 0.1679 & 1 & 0.4072 & 0.7908 & 0.0052 & -M \\ 0.4825 & 0.5767 & 0 & 0.1124 & 0.4621 & 0 \\ 0 & 0 & -M & 1 & 1 & 1 \end{pmatrix} \tag{60}$$

4.2.3. Calculation of the Disappointment Value and Elation Value of Both Parties

After obtaining the subjective satisfaction matrix of both parties, the disappointment value, elation value, and adjusted satisfaction of the matching objects in $P$ and $Q$ can be calculated. For each power user $P_i$ and electricity sales package $Q_j$, set the sensitivity parameters of disappointment $\alpha_i^P = \alpha_j^Q = 0.8$, the degree of aversion to disappointment to $\gamma_i^P = \gamma_j^Q = 0.5$, and the sensitivity parameters of elation to $\beta_i^P = \beta_j^Q = 0.8$, where $i = 1, 2, \ldots, 5$, $j = 1, 2, \ldots, 6$.

Taking $R^1$ as an example, the disappointment value, elation value, and adjusted satisfaction of the power user are calculated. According to the subjective satisfaction matrix, the potential object set $\theta^{P_1} = \{Q_3, Q_4, Q_5, Q_6\}$ in $Q$ that can match $P_1$ is derived. We can get

$$prob(P_1, Q_1) = prob(P_1, Q_2) = 0,$$
$$prob(P_1, Q_3) = prob(P_1, Q_4) = prob(P_1, Q_5) = prob(P_1, Q_6) = \tfrac{1}{4}$$

It is known from the above results that $P_1$ cannot match $Q_1$ and $Q_2$, i.e., $\overline{u_{11}^P} = \overline{u_{12}^P} = 1$. If $P_1$ matches $Q_3$ not $Q_4$, $P_1$ will be disappointed because $u_{13}^P < u_{14}^P$. If $P_1$ matches $Q_3$, the set of matched objects that cause $P_1$ to experience disappointment is $\Delta_{13}^{DP} = \{Q_4\}$, and the value of disappointment is $d_{13}^P = \tfrac{1}{4}\{1 - \exp((0.7692 - 0.6468)\ln 0.8)\} = 0.0029$. Similarly, if $P_1$ matches $Q_4$, $Q_5$ or $Q_6$, the disappointment value is

$$d_{14}^P = 0$$
$$d_{15}^P = \tfrac{1}{4}\{1 - \exp((0.6078 - 0.1708)\ln 0.8) + 1 - \exp((0.7592 - 0.1708)\ln 0.8)\} = 0.0248$$
$$d_{16}^P = \tfrac{1}{4}\{1 - \exp((0.6468 - 0)\ln 0.8)) + (1 - \exp((0.7692 - 0)\ln 0.8)) + (1 - \exp((0.1818 - 0)\ln 0.8))\} = 0.0376.$$

In addition, if $Q_3$ matches $P_1$, but not with $Q_5$ or $Q_6$, $P_1$ will feel happy because $u_{13}^p > u_{16}^p$, $u_{13}^p > u_{15}^p$. Therefore, if $Q_3$ matches $P_1$, the set of matching objects that make $P_1$ feel happy is $\Delta_{13}^{EP} = \{Q_5, Q_6\}$; then, the elation value is

$$e_{13}^P = \frac{1}{4}\{(0.5 - 0.5\exp((0.6468 - 0.1818)\ln 0.8)) + 0.5 - 0.5\exp((0.6468 - 0)\ln 0.8)) = 0.0131.$$

Similarly, if $P_1$ matches $Q_4$, $Q_5$, or $Q_6$, $P_1$'s happy value is

$$e_{14}^p = \tfrac{1}{4}\{0.5 - 0.5\exp((0.7692 - 0.6468)\ln 0.8) + 0.5 - 0.5\exp((0.7692 - 0.1818)\ln 0.8) + 0.5 - 0.5\exp((0.7692 - 0)\ln 0.8) = 0.0174$$
$$e_{15}^P = \tfrac{1}{4}\{(0.5 - 0.5\exp((0.1818 - 0)\ln 0.8))\} = 0.0022$$
$$e_{16}^P = 0.$$

Further, the $P_1$ satisfaction of other objects after adjustment is calculated.

$$\overline{u_{13}^P} = 0.6570, \overline{u_{14}^P} = 0.7866, \overline{u_{15}^P} = 0.1556, \overline{u_{16}^P} = -0.0376$$

Similarly, the disappointment value $d_{ij}^P$ and elation value $e_{ij}^P$ ($i = 1, 2 \ldots 5, j = 1, 2 \ldots 6$) of each power user for each electricity sales package can be calculated. Set $D^P = \left(d_{ij}^P\right)_{5 \times 6}$, $E^P = \left(e_{ij}^P\right)_{5 \times 6}$,

$$D^P = \begin{pmatrix} - & - & 0.0029 & 0 & 0.0248 & 0.0376 \\ 0.0066 & 0.0031 & 0 & - & - & 0.0375 \\ 0.0097 & 0.0361 & - & 0 & 0.0032 & - \\ 0 & - & 0.0064 & 0.0444 & 0.0040 & 0.0114 \\ 0 & 0.0343 & - & 0.0329 & 0.0089 & 0.0058 \end{pmatrix} \tag{61}$$

$$E^P = \begin{pmatrix} - & - & 0.0131 & 0.0174 & 0.0022 & 0 \\ 0 & 0.0025 & 0.0650 & - & - & 0.0071 \\ 0 & 0.0090 & - & 0.0611 & 0.0043 & - \\ 0.0446 & - & 0.0064 & 0 & 0.0075 & 0.0135 \\ 0.0536 & 0 & - & 0.0005 & 0.0102 & 0.0275 \end{pmatrix} \tag{62}$$

The disappointment value $d_{ij}^Q$ and elation value $e_{ij}^Q$ ($i = 1, 2 \ldots 5, j = 1, 2 \ldots 6$) of each power-selling company's power-selling package to electric power customers are calculated:

$$D^Q = \begin{pmatrix} - & - & 0.0499 & 0.0969 & 0.0749 & 0.0180 \\ 0.0216 & 0.0599 & 0 & - & - & 0.0494 \\ 0.0168 & 0 & - & 0.0114 & 0.0740 & - \\ 0 & - & 0.0863 & 0.0796 & 0.0282 & 0.1039 \\ 0.0431 & 0.0750 & - & 0 & 0 & 0 \end{pmatrix} \tag{63}$$

$$E^Q = \begin{pmatrix} - & - & 0.0068 & 0.0212 & 0.0212 & 0.0184 \\ 0.0240 & 0.0235 & 0 & - & - & 0.0047 \\ 0.0020 & 0 & - & 0.0210 & 0.0008 & - \\ 0.0107 & - & 0 & 0.0018 & 0.0101 & 0 \\ 0 & 0 & - & 0.0531 & 0.0640 & 0.0506 \end{pmatrix} \tag{64}$$

### 4.2.4. Adjustment of the Satisfaction Matrix

According to the disappointment value matrix, the elation value matrix, and the satisfaction matrix, the satisfaction matrix of the adjusted electricity customers and the electricity selling companies is calculated as follows:

$$\overline{U^P} = \left(\overline{u_{ij}^P}\right)_{5 \times 6} = \begin{pmatrix} -M & 1 & 0.6570 & 0.7866 & 0.1592 & -0.0376 \\ 0.0548 & 0.1329 & 1.0650 & 0 & -M & 0.2324 \\ 0.0741 & 0.3248 & -M & 1.0611 & 0.2221 & 0 \\ 1.0446 & -M & 0.5410 & -0.0444 & 0.5939 & 0.6570 \\ 1.0536 & -0.0343 & 0.3656 & -0.0092 & 0.4878 & 0.9789 \end{pmatrix} \tag{65}$$

$$\overline{U^Q} = \left(\overline{u_{ij}^Q}\right)_{5\times6} = \begin{pmatrix} 1 & -M & 0.2394 & -0.0969 & -0.0749 & 0.6157 \\ 0.1268 & 0.0956 & 1.0282 & -M & 0.9421 & 0.2783 \\ 0.1531 & 1.0237 & 0.4072 & 0.8004 & -0.0680 & -M \\ 0.4932 & 0.5767 & -0.0863 & 0.0346 & 0.4440 & -0.1039 \\ -0.0431 & -0.0750 & -M & 1.0531 & 1.0640 & 1.0506 \end{pmatrix} \quad (66)$$

### 4.2.5. Construction of a Stable TSMDM Model

On the basis of the satisfaction matrix $\overline{U^P}$ and $\overline{U^Q}$, a two-objective optimization model is established with the goal of maximizing the overall satisfaction of both parties:

$$\begin{aligned} & \max Z_1 = \sum_{i=1}^{5} \sum_{j=1}^{6} \overline{u_{ij}^P} x_{ij} \\ & \max Z_2 = \sum_{i=1}^{5} \sum_{j=1}^{6} \overline{u_{ij}^Q} x_{ij} \\ & \sum_{j=1}^{6} x_{ij} \leq 1, \ i = 1, 2, \ldots, 5 \\ & \sum_{i=1}^{5} x_{ij} \leq 1, \ j = 1, 2, \ldots, 6 \\ & \sum_{\overline{u_{ih}^P} > \overline{u_{ij}^P}} x_{ih} + \sum_{\overline{u_{kj}^Q} > \overline{u_{ij}^Q}} x_{kj} + x_{ij} \geq 1, \ \forall (P_i, Q_j) \in \Omega \\ & x_{ij} \in \{0, 1\}, \ i = 1, 2, \ldots, 5, \ j = 1, 2, \ldots, 6 \end{aligned} \quad (67)$$

Let $\omega = 0.5$; then, convert the two-objective optimization model into a single-objective optimization model, as follows:

$$\begin{aligned} & \max Z = 0.5 \sum_{i=1}^{5} \sum_{j=1}^{6} \overline{u_{ij}^P} x_{ij} + 0.5 \sum_{i=1}^{5} \sum_{j=1}^{6} \overline{u_{ij}^Q} x_{ij} \\ & \sum_{j=1}^{6} x_{ij} \leq 1, \ i = 1, 2, \ldots, 5 \\ & \sum_{i=1}^{5} x_{ij} \leq 1, \ j = 1, 2, \ldots, 6 \\ & \sum_{\overline{u_{ih}^P} > \overline{u_{ij}^P}} x_{ih} + \sum_{\frac{u_{kj}^Q}{Q} > \overline{u_{ij}^Q}} x_{kj} + x_{ij} \geq 1, \ \forall (P_i, Q_j) \in \Omega \\ & x_{ij} \in \{0, 1\}, \ i = 1, 2, \ldots, 5, \ j = 1, 2, \ldots, 6? \end{aligned} \quad (68)$$

The optimal solution of model (68) is $X^* = \left(x_{ij}^*\right)_{5\times6}$, expressed by the matrix

$$X^* = \begin{pmatrix} 0 & 0 & 0 & 0 & 1 & 0 \\ 0 & 0 & 1 & 0 & 0 & 0 \\ 0 & 0 & 0 & 1 & 0 & 0 \\ 1 & 0 & 0 & 0 & 0 & 0 \\ 0 & 0 & 0 & 0 & 0 & 1 \end{pmatrix} \quad (69)$$

Using the above matrix, the optimal matching results between five electric power customers and six power sales packages are obtained: $\mu = \{(P_1, Q_5), (P_2, Q_3), (P_3, Q_4), (P_4, Q_1), (P_5, Q_6)\}$. Thus, no electric power customers match power sales packages $Q_2$.

## 5. Conclusions

In this paper, a two-sided matching decision-making method based on disappointment theory was proposed. Firstly, this method is based on incomplete fuzzy preference relationships, which expands the applicability of recommendations. Secondly, by incorporating the influence of disappointment theory, the accuracy and efficiency of recommendations are improved. Lastly, the goal to maximize the power-users' overall satisfaction with proper choices of electricity sales packages is achieved, which can ensure the optimality and stability of recommendation results. The advantages of the methods proposed are as follows: firstly, the method proposed in the article is based on the incomplete fuzzy preference relationship between power users and electricity sales packages, which can overcome the power users' problem with lacking preference data caused by limited knowledge and different cultural backgrounds; secondly, the two-sided matching decision-making method proposed in this article also incorporates the influence of disappointment theory, which can not only better measure the satisfaction of power users with electricity sales packages, but also help set decision-making goals and greatly improve decision-making efficiency. In addition, by comparing the preference information of two matching objects one by one, this method can greatly reduce the burden in the process of extracting preference information, extract preference information faster and more flexibly, and ensure good recommendation results. At the same time, this method can also promote the development of a low-carbon economy under the background of "dual carbon" landing.

However, the bilateral matching decision-making method proposed in this article also has certain limitations. Firstly, the method was put forward only on the basis of the one-to-one problem without considering one-to-many matching problems. Secondly, the method only applies to the accurate evaluation of electricity sales packages by power users instead of further discussing the fuzzy preference relationships or hesitant preference information that may exist in real life.

Therefore, the focus of future research will be on addressing the above two shortcomings; the consensus issues and complex preference structure issues of power users will be deeply considered, and further in-depth research on the matching problem between power users and electricity sales packages will be conducted to more accurately and efficiently recommend electricity sales packages.

**Author Contributions:** Conceptualization, J.R. and Y.M.; methodology, Y.M.; software, Y.W.; validation, J.R., Y.W. and Y.M.; formal analysis, Y.M; investigation, Y.W.; resources, Y.M.; data curation, J.R.; writing—original draft preparation, J.R.; writing—review and editing, J.R.; visualization, J.R.; supervision, Y.M.; project administration, Y.M. All authors have read and agreed to the published version of the manuscript.

**Funding:** This work was supported by the National Natural Science Foundation of China (No. 72301248). This work was also supported by the Fundamental Research Funds of Zhejiang Sci-Tech University (No. 23222130-Y). This work was also supported by the Natural Science Foundation of Zhejiang Province (No. LQ22E070009).

**Institutional Review Board Statement:** Not applicable.

**Informed Consent Statement:** Not applicable.

**Data Availability Statement:** Data is available on request from the corresponding author.

**Conflicts of Interest:** The authors declare no conflict of interest.

## Nomenclature

| | |
|---|---|
| $P$ | Collection of electric power customers |
| $Q$ | Collection of electricity sales packages |
| $\mu$ | One-to-one mapping relationship |
| $u_{ij}^{P}$ | Subjective satisfaction of the $i$-th power user with the $j$-th electricity sales package |
| $u_{ij}^{Q}$ | Subjective satisfaction of the $j$-th electricity sales package with the $i$-th electricity user |
| $R^{i}$ | Incomplete fuzzy preference relationships provided by power user $i$ |
| $T^{j}$ | Incomplete fuzzy preference relationships provided by electricity sales package $j$ |
| $\omega^{P_i}$ | Priority weight vector for the power user $P_i$ |
| $\omega_{j}^{P_i}$ | Priority weight of the $i$-th user for the $j$-th type of electricity sales package |
| $\omega^{Q_j}$ | Priority weight vector of electricity sales package $Q_j$ |
| $\omega_{i}^{Q_j}$ | Priority weight of the $j$-th electricity sales package for the $i$-th electricity user |
| $\Delta^{P_i}$ | A set containing effective preference information of user $P_i$ |
| $\Delta^{Q_j}$ | A set containing effective preference information of the package $Q_j$ |
| $d_{ij}^{P}$ | The disappointment value of $P_i$ |
| $e_{ij}^{P}$ | The elation value of $P_i$ |
| $d_{ij}^{Q}$ | The disappointment value of $Q_j$ |
| $e_{ij}^{Q}$ | The elation value of $Q_j$ |
| $D_i(x)$ | The disappointment value function of $i$ |
| $E_i(x)$ | The elation value function of $i$ |
| $\alpha_{i}^{P}$ | The disappointment value sensitive parameter for the $i$-th user in set $P$ |
| $\beta_{i}^{P}$ | The elation value sensitive parameter for the $i$-th user in set $P$ |
| $\overline{U^{P}}$ | The satisfaction of electricity user $P$ after adjustment |
| $\overline{U^{Q}}$ | The satisfaction of package $Q$ after adjustment |
| $\omega$ | Weight coefficient |

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
