# Peer review of "Two-Sided Matching Decision Method of Electricity Sales Package Based on Disappointment Theory"

_applsci, doi:10.3390/app13179683_

Round 1

Reviewer 1 Report

This paper has an interesting and prctical result on this field.

I woud suggest that the data that the authors have provided should 

be clearly defined and shown to support their research results.

Author Response

Point 1: I woud suggest that the data that the authors have provided should be clearly defined and shown to support their research results.

Response 1: Thank you very much for your feedback. This is indeed our mistake, as we did not clearly state the source of the data, which caused inconvenience for your reading. We explained the incomplete fuzzy preference data between 5 power users and 6 power sales packages: these data are based on an individual's preference level for all options, quantified by values between 0 and 1.The modifications have been highlighted in yellow,and it is shown as follows:

Five power users are required to provide their preference information over six electricity sales packages through pairwise comparisons by considering some criteria, including clean energy ratio, power quality, and power supply services .The incomplete fuzzy preference relationshipprovided by each power user is expressed by. At the same time, six electricity sales packages are also required to express their preference information over power users through pairwise comparisons considering their value, investment capability and so on. The incomplete fuzzy preference relationshipof each power sales package is recorded as , as follows:

Reviewer 2 Report

I think the paper needs some corrections.The manuscript investigates an interesting topic. However, some corrections are needed.  Firstly, please check that your manuscript complies with the journal rules.  The numbering of the word "Introduction" as zero and the lowercase letter i should be checked to determine whether it is included in the rules. "iSlect" is written in Line 120 and "Check24" is written in Line 121, but it does not make any sense. Line 164 " judgments" should be corrected. Under subheading 1.1.1 many information and definitions are presented but none of them are cited. This is a general problem in this manuscript. There are unsourced graphics and definitions throughout the article. It should be revised in terms of academic writing rules.  In Experiment 15 matrix 1 column 2 row 2, both j=2 and j is not equal to 2. 

The manuscript is based on Disappointment Theory developed in Bell (1985), but there is not enough information about Bell (1985).

Also, since Bell (1985) does not include fuzzy preferences, the mathematical relationship between the method in the article and Bell (1985) should be explained more clearly. Finally, detailed information should be given about other papers (main ones) that use the theory in Bell (1985) (https://www.jstor.org/stable/170863) and their methods. 

Author Response

Because the formula  typed from mathtype cannot be displayed. Please see the attachment. I am very sorry for the inconvenience caused to you. Please understand. Thank you!

Reviewer 3 Report

The English language phrasing must be improved.

Major comments:

1.)  Loosely speaking, the model in the paper has three main components that contribute to its novelty:  fuzzy prefrences, disappointment theory, and matching.  Though (some of) the advantages of the first two are presented in the introduction, as well as some of the disadvantages of current approaches, it is not clear why these new features add to the specific context of electricity.

2.)  The authors should add a general explanation of how and why fuzzy preferences and disappointment theory combine to generate preferences on the two sides of the market before launching into the technicalities of the evaluation processes.  As written, the link between sections 1 and 2 is not as clear as it should be. 

Minor comments:

-Section 0, ``introduction'' should be capitalized.

-Are the opening sentences of the first and second paragraphs referring to the power sector of society in general, or China more specifically? Based on the references it seems the latter, but at times the phrasing makes it seem intended as the former, and the context matters for the application.

-Page 6, line 258:  Is the ``h'' before ``h harmonic'' a typo or is it referring to the previously mentioned subscript?

-Page 22, lines 741-742:  No spaces before or after 1. or 2.

The English language phrasing must be improved.

Author Response

Point 1: The English language phrasing must be improved.

Response 1: Thank you very much for your question.We have made revisions to the expressions of nouns in the article to make them more appropriate in response to your suggestion. Secondly, we have also refined the expression of certain sentences to become more concise. For example:

  • In the abstract, “In the context of "dual carbon", with the continuous deepening of the power system reform, the application of high proportion renewable energy is becoming increasingly widespread.” is changed to “Under the background of the "dual carbon" targets and continuously promoted power system reform, the application of high proportion renewable energy is becoming increasingly widespread.”
  • In the abstract, “All sectors of society have put forward higher requirements for electricity sales packages, which can guide the behavior of electric power customers, promote energy conservation and emission reduction, and finally promote low-carbon operation of the power market economy.” is changed to “All sectors of society have greater demands for more appropriate electricity sales packages to guide the behavior of power users, which will in turn help conserve energy, reduce emission, and finally achieve low-carbon operation of the power market economy.”
  • In the abstract, we delete the “Therefore, the requirements of electric power customersand power-selling companies for the power-selling package are increasing, which puts forward higher requirements for the power-selling package recommended by the power-selling company.” because this sentence has the same meaning as the previous sentence in the abstract.
  • In the abstract, “However, the existing recommendation methods are unable to make appropriate recommendations for the users with a lack of preference information.” is changed to “However, the existing methods of recommending electricity sales packagesfail to provide appropriate and accurate recommendations for the users lacking preference information.”
  • In the conclusion, “the goal is to maximize the overall satisfaction between electric power customersand electricity sales packages,” is changed to “the goal to maximize the power-users’ overall satisfaction with proper choices of electricity sales packages is achieved.”
  • In the conclusion, “which can overcome the problem of missing preference data caused by different knowledge and cultural backgrounds of electric power customers,” is changed to “which can overcome the power users’problem with lacking preference data caused by limited knowledge and different cultural backgrounds .”
  • In the conclusion, “The method only considers the one-to-one problem but didn’t consider the one-to-many matching problems.,” is changed to “The method is put forward only based onthe one-to-one problem without considering the one-to-many matching problems .”
  • In the last paragraph of the conclusion, we upgraded the language expression by improving “Therefore, in future research, the focus of research will be to address the above two shortcomings: the consensus issues and complex preference structure issues of electric power customers will be deeply considered, and further in-depth research will be conducted on the matching problem between electric power customers and electricity sales packages to more accurately and efficiently recommend electricity sales packages.”to “Therefore, the focus of future relative researches will be on addressing the above two shortcomings: the consensus issues and complex preference structure issues of power users will be deeply considered, and further in-depth researches on the matching problem between power users and electricity sales packages will be conducted to more accurately and efficiently recommend electricity sales packages.” 
  • In the first paragraph of section 3.4, “He believed that ”is changed to “It is argued that”.”The two-way choice between power users and electricity sales packages is the product of satisfying two-sided satisfaction at the same time” was changed to “The two-way choice between power users and power sales packages is a product that satisfies both sides of satisfaction.”
  • In the entire manuscript, we replace the expression of “joy” with “elation”. Because the degree of “elation” is heavier than “joy”, it can better depict the emotions of both matching parties.
  • In the entire manuscript, we replace “power selling”with “power-selling”, because power-selling company is an integral part. So it needs “-” to connect “power” and “selling”.
  • In the entire manuscript, we replace “power users”with “electric power customers”, because the meaning of “power users” is too general, while “electric power customers” effectively characterizes the applicable group.
  • In the second paragraph of the first section, we change "extracting customer behavior or load curves from the time series, in order to more targeted recommended electricity sales packages to users." to "extracting customer behavior or load curves from the time series to more targeted recommended
  • Change "in Spark" in line 103to "in the Spark".
  • Change "method of" in line 133 to "method for".
  • Delete the "," after "clustering" in line 112.

There are still a large number of revisions that are not listed here. The revisions are highlighted in yellow in the manuscript.

Point 2:  Loosely speaking, the model in the paper has three main components that contribute to its novelty:  fuzzy prefrences, disappointment theory, and matching.  Though (some of) the advantages of the first two are presented in the introduction, as well as some of the disadvantages of current approaches, it is not clear why these new features add to the specific context of electricity.

Response 2: Thank you very much for your question. As the decision-making method proposed in this article is mainly used to obtain the optimal matching relationship between power users and electricity sales packages, in order to recommend the most suitable electricity sales package to power users. In order to be closely related to the theme and not deviate from it, the background related to electricity was included in the method explanation.

Point 3:   The authors should add a general explanation of how and why fuzzy preferences and disappointment theory combine to generate preferences on the two sides of the market before launching into the technicalities of the evaluation processes.  As written, the link between sections 1 and 2 is not as clear as it should be. 

Response 3: Thank you very much for your question. In the electricity market, users usually have limited understanding of electricity sales packages, and the knowledge and information they possess are also limited. In this case, users often can only provide a vague general preference relationship, but cannot accurately express detailed preferences. For example, they might indicate a preference for a certain attribute as "high" or "moderate" rather than giving a specific numerical value. This fuzzy preference relationship can better reflect the cognitive limitations of users on electricity sales packages. Furthermore, disappointment theory also plays a key role in this context. In the Introduction, we have explained the concept of disappointment theory that users have emotional experiences when faced with choices that may lead to disappointment. Users may have expectations for certain electricity sales packages, and they may be disappointed if these expectations cannot be realized. Conversely, they may be delighted if certain packages exceed their expectations. This psychological experience of disappointment and joy is one of the important driving factors behind user preferences. Combining fuzzy preference relations with disappointment theory can more accurately characterize user satisfaction. In the electricity market, due to the large number of packages, users are faced with the difficulty of making choices. The introduction of the bilateral matching method allows the system to comprehensively consider the user's fuzzy preference relationship and disappointment and joy psychology, so as to provide users with the best package matching. In this way, users do not need to face a complicated selection process, but can get recommendations for electricity sales packages that meet their vague preferences and disappointment and joy, thereby improving their satisfaction.

 The above additions are highlighted in yellow in the manuscript.

Point 4: -Section 0, ``introduction'' should be capitalized.

Response 4: Thank you for your meticulous review work and for discovering such a low-level writing mistake. I have made corrections in the manuscript: change “introduction” to “Introduction”, which has been highlighted in yellow.

Point 5: Are the opening sentences of the first and second paragraphs referring to the power sector of society in general, or China more specifically? Based on the references it seems the latter, but at times the phrasing makes it seem intended as the former, and the context matters for the application.

Response 5: Thank you very much for your question.The first paragraph of the article draws out the matching problem between power users and power sales packages under the background of "dual carbon" implementation. Not only China, but also the whole world are currently facing the double carbon problem. In order to avoid ambiguity, I add the following sentence in the 64th line of the manuscript: “Not only China, but the whole world is facing the double carbon problem. EU member states are currently focusing on it Renewable Energy and Energy Efficiency Program [1].”  Change "A large number of power selling companies have emerged" to "A large number of power selling companies have emerged around the world". The above additions are highlighted in yellow in the manuscript.

[1]D. C. Momete, Analysis of the Potential of Clean Energy Deployment in the European Union, IEEE Access, 2018 ,6,54811-54822,

Point 6: Page 6, line 258:  Is the ``h'' before ``h harmonic'' a typo or is it referring to the previously mentioned subscript?

Response 6: Thank you for your meticulous review work and for discovering such a low-level writing mistake. I have made corrections in the manuscript: delete the “h” before “harmonic”, which has been highlighted in yellow.

Point 7: Page 22, lines 741-742:  No spaces before or after 1. or 2.

Response 7: Thank you for your meticulous review work and discovering formatting issues in the manuscript. This is indeed a mistake in my writing, and I have corrected it, which has been highlighted in yellow.

Round 2

Reviewer 2 Report

The authors appear to have made the corrections provided by me. If some minor errors are corrected, it will be a better article.  However, the matrix representations are very messy. Also, there are still no references under the figures. It would be more understandable if the equations between lines 411-425 were shown one below the other instead of side by side.  There is a three-step method between line 472 and line 480. Which study is the source of this method?  In Equation 46, there are symbols under the sum symbol. This is the first time I have seen this type of notation. What is the meaning of these symbols ? Could authors explain them by referring to a math book or a statistics or topology book?

I wish you success.

Author Response

Dear reviewer, as my reply contains a formula from MathType and cannot be displayed in the text box. So please see the attachment.

Reviewer 3 Report

Just a few very minor comments (typos).

-p.1: It may be due to spacing issues with both subscripts and superscripts involved, but the terms for ``the disappointed value of Q_j'' and ``the elation value of Q_j'' are offset in the notation list before the paper's introduction.

-p. 2, line 56: ``...is is being promoted further,'' (repeated instance of ``is'').

-p. 11, line 502: ``outcome ,which is known as disappointment aversion.'' (comma placement)

-p. 22, line 764, ``the focus of future relative researches,'' seems like it should be, "the focus of future research...'' Similarly, on line 764 ``research'' should also be singular rather than plural.

Not a suggestion, but just to let the authors know, I found the schematic diagram to be helpful.

The writing is not perfect (see minor comments above), but the paper does make more sense now. 

Author Response

Point 1: : It may be due to spacing issues with both subscripts and superscripts involved, but the terms for ``the disappointed value of Q_j'' and ``the elation value of Q_j'' are offset in the notation list before the paper's introduction.

Response 1: Thank you for your thorough review work. We have addressed the issue of positional misalignment for "the disappointed value of Q_j" and "the elation value of Q_j." The modifications have been highlighted in green within the text.

Point 2:  line 56: ``...is is being promoted further,'' (repeated instance of ``is'').

Response 2: Thank you for your meticulous review work. We have removed the extra "is" from the 56th line, where it originally said "is being promoted further." The modified sections in the text are highlighted in green.

Point 3:   line 502: ``outcome ,which is known as disappointment aversion.'' (comma placement)

Response 3: Thank you for your meticulous review work. We have removed the comma from the 502nd line, so it now reads "outcome which is known as disappointment aversion." This change makes the subordinate clause "which is known as disappointment aversion" a part of the main sentence, further explaining the content of the "expected outcome." The modified sections in the text are highlighted in green.

Point 4:  line 764, ``the focus of future relative researches,'' seems like it should be, "the focus of future research...'' Similarly, on line 764 ``research'' should also be singular rather than plural.

Response 4: Thank you for your meticulous review work. We have corrected "the focus of future relative researches" to "the focus of future research..." Additionally, we have changed "researches" to "research" in the 766th line. The modified sections in the text are highlighted in green.
